# Food Tracking Perspective: DNA Metabarcoding to Identify Plant Composition in Complex and Processed Food Products

**DOI:** 10.3390/genes10030248

**Published:** 2019-03-25

**Authors:** Antonia Bruno, Anna Sandionigi, Giulia Agostinetto, Lorenzo Bernabovi, Jessica Frigerio, Maurizio Casiraghi, Massimo Labra

**Affiliations:** 1Zooplantlab, Department of Biotechnology and Biosciences, University of Milano-Bicocca, Piazza della Scienza 2, I-20126 Milano, Italy; antonia.bruno@unimib.it (A.B.); g.agostinetto@campus.unimib.it (G.A.); andrea.galimberti@unimib.it (L.B.); maurizio.casiraghi@unimib.it (M.C.); massimo.labra@unimib.it (M.L.); 2FEM2-Ambiente, Piazza della Scienza 2, I-20126 Milano, Italy; jessica.frigerio@fem2ambiente.com

**Keywords:** DNA metabarcoding, *trn*L, molecular markers, High Throughput Sequencing, food tracking, food contaminants, processed food, food matrices, herbal products

## Abstract

One of the main goals of the quality control evaluation is to identify contaminants in raw material, or contamination after a food is processed and before it is placed on the market. During the treatment processes, contamination, both accidental and economically motivated, can generate incongruence between declared and real composition. In our study, we evaluated if DNA metabarcoding is a suitable tool for unveiling the composition of processed food, when it contains small trace amounts. We tested this method on different types of commercial plant products by using *tnr*L marker and we applied amplicon-based high-throughput sequencing techniques to identify plant components in different food products. Our results showed that DNA metabarcoding can be an effective approach for food traceability in different type of processed food. Indeed, the vast majority of our samples, we identified the species composition as the labels reported. Although some critical issues still exist, mostly deriving from the starting composition (i.e., variable complexity in taxa composition) of the sample itself and the different processing level (i.e., high or low DNA degradation), our data confirmed the potential of the DNA metabarcoding approach also in quantitative analyses for food composition quality control.

## 1. Introduction

One of the main goals of food quality control is to identify contaminants or counterfeits in raw material, during food processing and before market placement [1]. During treatment processes, contamination, both accidental or economically motivated, can lead to incongruences between declared and real composition of the final food products. These incongruences could be threats to consumers health and may reduce potential health benefits of the products, even causing allergic or toxic reactions [2]. A faster, more efficient and cost-effective quality check can solve these issues and promotes the import and export of verified food products to different countries [3,4]. Together with the production of novel foods, methods for the development of new products and new production strategies (i.e., the choice of ingredients or components, the design of food processing, shelf-life studies and sensory evaluation of products) are steadily changing equilibria between the two forces. Consequently, the analysis and control of food content are constantly being developed. After recent technological advanced, molecular techniques, based on the amplification and/or sequencing of marker DNA regions, can be a useful diagnostic method to identify food species composition and represents one of the the most promising tools [5].

The use of DNA-based markers in molecular authentication allows accurate and sensitive results, even in cases of phylogenetically related species [6]. This is proved by the collaboration among FDA’s Center for Food Safety and Applied Nutrition (CFSAN) and Center for Veterinary Medicine (CVM), the University of Guelph’s Biodiversity Institute of Ontario, Canadian Center for DNA Barcoding, and the Laboratories of Analytical Biology at the Smithsonian National Museum of Natural History, Suitland, MD, USA. A coordinated network of these institutes resulted in a universal and standardized approach for metazoan identification based on DNA barcoding: they first provided details on protocols, reagents, and equipment required to carry out a validation study for barcode generation for fish identification in the form of FDA Laboratory Information Bulletin No. 4420. Although the molecular approach is well supported and validated in the case of the identification of a single biological raw material [7,8,9,10], other issues arise when dealing with treated or adulterated foods. In fact, highly processed products are one of the challenges of food traceability because DNA could undergo degradation processes due to the intense physico-chemical conditions of industrial treatments. As a consequence, in several processed foods only very short fragments of DNA are available for the analysis and typically the common DNA barcoding analysis fails. In this context, High-Throughput DNA Sequencing (HTS) techniques (also called Next Generation DNA Sequencing, NGS), combined with powerful tools for data analysis, are revolutionizing food quality control. HTS can be applied using an amplicons-based strategy (known as DNA metabarcoding) or a genomes-based strategy (known as metagenomics). In particular, DNA metabarcoding is presently a new standard in the quality control of new food products [11,12,13]. Nevertheless, DNA metabarcoding still has some limitations: (i) DNA fragmentation [14,15]; (ii) DNA amplification biases [16,17]; (iii) the real ‘primer universality’ [18]; (iv) the presence of DNA amplification inhibitors [19]; (v) the occurrence of (food) materials in trace and low DNA yield [17]; (vi) the accidental laboratory contamination when the target DNA is fragmented and of low concentration. A sequence variation due to these restrains hampers species detection in food matrices.In particular when a false mutation derived to degenerative process occurs, the assignment of the incorrect query sequence returns false positive due a misidentified assignment [20].

The intron fragment P6-loop (10–143 bp) of the chloroplast tRNALeu (UAA) intron sequences [*tnr*L (UAA), 254–767 bp] is a short, but informative sequence proved by Taberlet et al. [21] to be effective in identifying plant species in processed food and ancient permafrost samples. Despite closely related species are not always resolved [21], food industry [22], forensic sciences [23,24] and diet studies based on faeces [25] successfully used *tnr*L marker for plant identification and this contributed to increment the number of *tnr*L sequences present in databases. In our study, we evaluated if DNA metabarcoding is a suitable tool for unveiling the composition of processed food, even when it contains trace amounts. We tested this method on different types of commercial plant products by using the *tnr*L marker.

Additionally, we tested if it is possible to estimate the quantity of the original biological components. For this reason, we explored the correlation between food composition (biomass or DNA content) and the corresponding assigned reads: we set up, under controlled conditions, a mock herbal mixture known in composition, for which we commissioned a herbalist’s shop and simulated herbal products, and an artificial fruit mixture, replicated in different proportions, simulating exotic fruit juices. Our results showed that DNA metabarcoding is an effective tool for food traceability in different kinds of processed food. Indeed, in the vast majority of our samples, we identified the species composition as the labels reported. However, the universality of this approach is affected by food composition (i.e number of species and their amount) and the level of processing (i.e., DNA quality). The tests on mock mixtures evidenced a good relationship between food composition and number of DNA reads obtained and this confirms the power of DNA metabarcoding also in quantitative analyses.

## 2. Materials and Methods

### 2.1. Commercial Processed Foods and Mock Herbal Mixture for Qualitative Analysis

To test the efficacy of DNA metabarcoding, we chose six different commercial products (described in Figure 1) with the following characteristics: (i) ingredients not easily distinguishable by morphology; (ii) specified ingredients in the label; (iii) different processes involved in the production (i.e., deep-freezing, lyophilization). As representative of these features, we selected from the market processed food such as vegetable stock cube, curry, pureed soup of deep-frozen vegetables, or food supplements, that are characterized by high levels of food processing and, at the same time, high level of complexity considering taxa composition. In particular, considering processing, vegetable stock cube undergoes to high temperature processing and dehydration. Curry is a complex combination of spices or herbs, usually including ground turmeric, cumin, coriander, ginger, processed in the form of powder. Pureed soup of deep-frozen vegetables undergoes both to high temperature processing and deep-freezing for storage. Food supplements can be manufactured using intact sources, phytoextracts or other not specified fractions from plants, and can be in the form of liquid, pills, capsules or tablets, as in our case study. We also selected products that are not characterized by a high level of complexity considering taxa composition, but can be subjected to adulteration, such as flavoured tea and saffron. In particular, saffron adulteration often occurs due to the high cost of the raw material. Commonly, adulteration consist in adding or mixing of similar materials such as beet, pomegranate, red-dyed silk fibers, safflower, and marigold or honey and vegetable oils [26]. Moreover, a mixture of plants, mimicking commercial herbal products, known in composition and quantity (100 mg in dry weight for each specimen), was created by a herbalist’s shop.

### 2.2. Fruit Mixtures Known in Composition and Quantity for Quantitative Analysis

To test the efficacy of DNA metabarcoding analysis in identifying the composition of commercial products at the qualitative and quantitative level, we prepared five mixtures composed by five plants commonly used to prepare fruit juice and with consistent phylogenetic distances: Ananas (*Ananas comosus* Bromeliaceae), Avocado (*Persea americana* Lauraceae), Dragon fruit (*Hylocereus undatus* Cactaceae), Mango (*Mangifera indica* Anacardiaceae), Papaya (*Carica papaya* Caricaceae). To avoid possible laboratory contaminants, fruit not present in the ingredients declared in the selected commercial food and never processed in our laboratory were chosen. To avoid possible intra-species sequence variability, we used only one fruit of each type to create the artificial mixtures. The five mixtures were prepared by mixing different amounts of DNA of each species (Appendix A), in different combinations: in detail, DNA was individually extracted from each fruit and individually quantified using qPCR, as described in Section 2.3 and in [27]. Then, we prepared different dilutions of each DNA extract. Finally, we composed the artificial mixtures starting from the DNA extracts and combining equal volumes but different concentrations of DNA, as reported in Appendix A. As a consequence, we obtained five mixtures all composed by the same five fruits, but the combination of DNA fruit concentrations is peculiar of each mixture (Appendix A). The combination of different DNA quantities instead of fruit sample quantities expressed as weight was preferred in order to evaluate whether the number of features obtained is proportional to the DNA copy numbers of amplified *trn*L fragments.

### 2.3. DNA Extraction and qPCR

DNA extractions for commercial processed foods, mock herbal mixture and fruit used for artificial fruit mixtures were carried out using EuroGold Plant DNA Mini kit (Euroclone, Pero, Italy), following the manufacturer’s instructions. In detail, we started from 50 mg of sample material, that was homogenized via a mortar and liquid nitrogen; after lysis and wash steps, DNA was eluted in 100 μL of elution buffer. Three replicates of DNA extraction were created and then pooled together before further steps.

qPCR assays were performed with AB 7500 (Applied Biosystems, Carlsbad, CA, USA) on DNA extracts deriving from commercial processed foods, mock herbal mixture, each single fruit used for the artificial fruit mixtures and the artificial fruit mixtures themselves. qPCR conditions included an initial denaturation at $95{\;}^{\circ }C$ for 10${}^{\prime }$, followed by 40 cycles of denaturation at $95{\;}^{\circ }C$ for 15${}^{\prime \prime }$ and annealing-elongation for 1${}^{\prime }$ at $55{\;}^{\circ }C$. Real-time PCR was set up with 2X SsoFast EvaGreen Supermix with Low ROX (Bio-Rad S.r.l., Segrate (MI), Italy) in which EvaGreen was used as a detecting dye; a 10 μL reaction consisted of 5.0 μL SsoFast EvaGreen Supermix with Low ROX, 0.1 μL each 10 μmol/L primer solution, 2 μL DNA sample, and 2.8 μL of Milli-Q water. Primer sequences are given in Table 1. All samples and negative controls were run in triplicate. Amplification data were collected and analyzed with the SDS 7500 Real-Time PCR System Software (Applied Biosystems). All the procedures were conducted in the laminar flow cabinet, to avoid contamination with exogenous DNA and inter-samples contamination and in separate rooms for the pre and post amplification steps, with dedicated personal protective equipment (PPE).

### 2.4. Sanger Sequencing

The *trn*L region amplified by the primer pairs *trn*L g-h (the forward primer position is upstream from *trn*L c) [21] was sequenced in the samples belonging to exotic fruit used for artificial fruit mixtures, in order to obtain the reference sequences implementing our local reference database. PCR amplification was performed using KAPA HiFi HotStart Ready Mix (Kapa Biosystems, Woburn, MA, USA) in a 25 μL reaction volume, according to the manufacturer’s instructions. PCR cycles consisted of an initial denaturation for 10 min at $94{\;}^{\circ }C$, 35 cycles of denaturation (30${}^{\prime \prime }$ at $94{\;}^{\circ }C$), annealing (30${}^{\prime \prime }$ at $55{\;}^{\circ }C$), elongation (60${}^{\prime \prime }$ at $72{\;}^{\circ }C$), and final elongation at $72{\;}^{\circ }C$ for 6${}^{\prime }$. Primer sequences are given in Table 1. PCR products were purified from agarose gel with QIAquick Gel Extraction Kit (Qiagen, Hilden, Germany), following manufacturer’s protocol. Products were submitted for sequencing to Eurofins Genomics (https://www.eurofinsgenomics.eu/). The DNA strands were bidirectionally sequenced. All obtained sequences were manually edited, aligned, and used as query in an individual Basic Local Alignment Search Tool (BLAST) in GenBank to verify samples identity.

### 2.5. Libraries Preparation and Sequencing

Processed foods, mock herbal mixtures, and eight replicates for each artificial fruit mixture were tested. DNA extracts were normalized on Ct values of qPCR with the same primer pairs, instead of measuring the total amount of DNA with fluorometric/spectrophotometric methods. Libraries were generated following the MetaFast protocol (www.fasteris.com/metafast). For DNA amplifications, we used a primer pair (*trn*L g-h) targeting a short but informative fragment of *trn*L gene, the P6 loop region. Amplification conditions were the following: 2× KAPA HiFi HotStart Ready Mix (Kapa Biosystems) 12.5 μL, Forward Barcoded Primer [1 μM] 5 μL, Reverse Barcoded Primer [1 μM] 5 μL, 2.5 μL of DNA extract, in a final volume of 25 μL. Barcode sequences are listed in Appendix A. The PCR cycling parameters were 10${}^{\prime }$ at $94{\;}^{\circ }C$, and then 35 cycles with denaturation for 30${}^{\prime \prime }$ at $94{\;}^{\circ }C$, annealing for 30${}^{\prime \prime }$ at $55{\;}^{\circ }C$, elongation for 60${}^{\prime \prime }$ at $72{\;}^{\circ }C$, with a final extension for 6${}^{\prime }$. All PCR products were mixed together and purified with MinElute™ PCR purification kit (Qiagen, Hilden, Germany). The sequencing was carried out on the HiSeq 2500 sequencing platform (Illumina, San Diego, CA, USA) with a paired-end approach (2 × 125 bp). We introduced negative controls (no template) during the amplification step of library preparation. We checked the absence of amplification for the no template controls with QIAxcel capillary electrophoresis. DNA sequence obtained are available on EBI metagenomics database with following accession: ERS3208058-ERS3208104, Project: PRJEB31578.

### 2.6. Bioinformatics Analysis

DNA metabarcoding analysis of food composition was performed using the plugins of the QIIME2 suite [28]). Raw Illumina reads were paired and pre-processed using VSEARCH v2.5.0—merge pairs algorithm [29]. Reads were filtered out if ambiguous bases were detected and were shorter than 52 bp. Moreover, an expected error = 1 was used as an indicator of read accuracy [30]. Plant features were obtained using—cluster_fast algorithm with a 100% sequence identity with at least a depth of 500× for each feature. A random sequence was chosen as the representative sequence of the cluster. To decrease the false positive rate in the sequence population, a chimera detection analysis was performed on the obtained reference sequences. Since there is no reference database for *trn*L gene for chimera detection, we used uchime_denovo algorithm that carries out a de novo analysis without a reference. The taxonomic assignment of the representative sequences, to obtain the features, was carried out using the classify-consensus-vsearch plugin implemented in QIIME2 [31] against the local database adopting a consensus confidence threshold of 0.9. Reference feature sequences were aligned against our local reference database, built with downloaded *trn*L sequence available in NCBI at November 2018 and increased with Sanger sequences obtained in our laboratory for fruit mixture samples as a control. Where consensus confidence during classification was equal to 1 the query sequence was assigned to the corresponding species. An additional comparison with the non-redundant nucleotide NCBI database was performed in order to check the features that were not assigned by our reference database. To estimate differences among the replicates of each fruit mix, we performed beta diversity analysis. Bray-curtis distance was calculated and one-way PERMANOVA pairwise test was performed using beta-group-significance plugin implemented in QIIME2 platform [28]. To assess the accuracy with which the expected taxonomic composition is reconstructed after experimental analysis, we used the q2-quality-control plugin implemented in QIIME2. Linear regression scores between expected and observed feature abundances (qPCR copies vs reads amount) are calculated at species rank, and plots of accuracy and observation correlations are plotted.

## 3. Results and Discussion

### 3.1. Sequence Analysis

To asses the food composition in the six selected processed foods (*n* = 6) and in forty-one mock admixtures (*n* = 41), 6,308,509 quality-filtered *trn*L sequences were obtained. After filtering and primer removal, the remaining sequences were of high quality and had an average length of 75 bp (ranging from 32 to 84 bp) and clustered into 671 features (100% identity).

### 3.2. DNA Metabarcoding for Food Traceability

We focused on the DNA metabarcoding analysis of the six commercial products to estimate their composition and to compare it with the list of ingredients accompanying the selected products. The biological diversity observed in the commercial food is characterized by 22 taxa assigned, 16 of which at species level. If we assume the declared composition on the label as the true composition, the rate of false positive (misclassified features or real contaminants) occurred (*n* = 9) is lower than the false negative (not sequenced features) (*n* = 18). Given the different type of food processing and consequent different level of DNA degradation occurred for each product, we now discuss case by case the results obtained, resumed in Table 1 and illustrated in Figure 1.

A barchart in Figure 2 shows the composition of the six commercial products selected from the market and one mock herbal mixture as control.

Products such as saffron showed DNA of only declared species (see Table 2) and excluded any contamination, also during food processing. Flavoured tea returned 99.7% of *Camellia sinensis* features, without the presence of the flavoured declared. This is probably due to the fact that only the aromatic compounds are present as chemical flavours and not the DNA. A peculiar case is represented by the food supplement, where we obtained the almost totality of sequences assigned to *Taraxacum officinale*, despite the highly complex composition declared in the ingredient label. There could be different explanations. First, the non correspondence in what is declared and what is really used: *T. officinale* is a common and low cost species largely used in several food supplement. We cannot exclude the occurrence in low amount of the other species and another possible hypothesis is the presence of highly degraded DNA due to the processing of single herbal products. Concerning the most heterogeneous analyzed food, such as samples vegetable stock cube, curry, and deep frozen vegetables, our data reported a mixed composition reflecting, at least in part, the complexity of the food products. Moreover, DNA metabarcoding results suggested that in most cases the declared species were detected. In some samples we also detected undeclared food elements. For example, in the case of vegetable stock cube, we found a nearly perfect match in what is declared and what we detected, but we also detected unexpectedly *Crocus sativus* (2.1%), *Laurus* sp. (0.5%), *Medicago sativa* (0.3%) and *Foeniculum vulgare* (43%): they were not specified in the ingredients label, unless they were intended as part of the general category “spices”, or they derived from food process contamination. On the other hand, curry composition was only partially unveiled by DNA metabarcoding: we found, as expected, *Trigonella foenum-graecum* (36%), *Brassica* sp. (33.3%), *Coriandrum sativum* (13.2%), *Allium* sp. (4%). We did not detect *Cuminum cyminum*, since the *trn*L sequence is not deposited in NCBI. We found, as not declared species, *Daucus carota* (13.4%), a low-cost vegetable that could be present as contaminant or adulterant. Despite the presence of *trn*L sequences in reference database, we did not detect *Piper nigrum*, *Cinnamomum verum, Curcuma longa, Syzygium aromaticum* (sequence deposited only for genus *Syzygium*), *Myristica fragrans* and *Capsicum sp.* It can be related both to the low amounts used in the composition of this food and difficulties in DNA extraction. In the specific case of curry we underlined that, in our analysis, we detected the lower amount of reads for this sample and this could also suggest that this kind of food could contain food additives, polysaccharides, polyphenolics, and secondary metabolites that can affect PCR amplification, causing failures in downstream steps, such as sequencing [19,32,33]. The pureed soup of deep-frozen vegetables is composed by pureed and deep-frozen vegetables, but we were able to detect half of the species declared. We did not report the presence of *Petroselinum crispum, Allium* sp. and *Ocimum basilicum*, probably because were used in low amounts (in the declared ingredient compositions are in the second half of the list). *Cucurbita* sp. was not detected with DNA metabarcoding. Interestingly, we detected high amounts of *Pisum sativum* (24.8%) and *Foeniculum vulgare* (19.8%), that are not declared in the food label, and probably rely in the food chain production. Then, we wondered if we did not detect some species due to their real absence in the processed food or due to a failure of the method. For this reason, we commissioned from a herbalist’s shop an ad hoc, known in composition, mock herbal mixture as an experimental control. In this case, we recovered the 100% species expected and no contaminants were identified, confirming the reliability of the metabarcoding protocol. As well as a control, we based on this mock mixture the exploration of the possible relationship between quantities of herbs and number of reads obtained. The quantity used for each species, expressed as grams of dry weight, is equally distributed among all taxa. However, DNA metabarcoding analysis produced different proportion of reads: only *Betula pendula* showed the expected frequency (12%), and, vice versa, for example, *Taraxacum officinale*, instead of resulting in 12% frequency, was 29.3%, suggesting to be preferred in primer annealing. Noteworthy, we observed the same behavior in the Food Supplement sample, considering the reads assigned to *Taraxacum officinale* (99.2%), compared to those assigned to the other taxa, and this could be explained also with primer competition effect. HTS is a powerful tool to detect various sources of DNA in a single analysis, trace quantity of foreign species DNA [34], but occurrence of PCR amplification bias [35] may cause inaccurate estimation of species composition and misleading results. Nevertheless, in the case of a mock herbal mixture, even if there was not a perfect linear relationship between weight and number of reads, the observed correlation coefficient outlined a certain degree of relationship between observed and expected frequencies (r = 0.75, *p*-value < 0.001). Our results showed that the sensitivity and accuracy of DNA metabarcoding makes it effectively a powerful tool for food authentication and to trace food contaminants. Nevertheless, DNA metabarcoding in plants does have limitations, mainly arising from the combination of difficulty in DNA amplification due to degraded DNA in the processed samples and the primer bias in the annealing step.

### 3.3. DNA Metabarcoding to Quantitatively Evaluate Food Composition

As already noted in our results on processed food (see in the previous paragraph the case of *Taraxacum officinale*), amplicon metabarcoding can be affected by the PCR amplification step using “universal” markers. The occurrence of a bias during PCR amplification may cause the inaccurate estimation of quantities and this was at least partially demonstrated for metazoan and plants [17,35,36]. This bias generates a variable number of template–primer mismatches across species, resulting in a final amplified DNA mixture that does not always reflect the original proportion of each species, limiting the quantitative potential of DNA metabarcoding [37,38,39,40]. Assuming probability of primer bias, we have been concerned about whether there is a correlation between DNA copies obtained through qPCR amplification and reads obtained by sequencing, for each tested species. This could shed a light on the evaluation of food composition also from a quantitative point of view, especially in the case of samples heterogeneous in composition. For this reason, we set up an assay where five different fruit mixtures were composed by five different exotic fruits, in different known proportions. We opted for fruit mixtures mimicking exotic fruit juices because fruit is relatively easy to authenticate by their morphological characteristics when intact and fresh, but give rise to the possibility of adulteration when processed into juice. In particular, adulteration may regard both composition and proportions of the fruit, preferring preferring cheaper and more available fruit. We decided to create our own ad hoc mixture without using any commercial fruit mixture, since in this way we could be sure that all the DNA sequences belonged to same fruit (for each fruit mixture), and there was no intra-species sequence variability. Fruit mixtures were analysed both with qPCR and Illumina HiSeq sequencing, using the same primer pairs. As shown in Figure 3, we were able to detect and correctly assign each exotic fruit used for mixture preparation, also thanks to our local reference database implemented with Sanger sequences generated for this experiment.

The replicates of each fruit mix showed a high repeatability of the obtained results, confirming the accuracy of metabarcoding method (The Bray-Curtis distance among each samples are reported in Appendix A). Analyzing the relationship between expected and observed quantities of fruit (Figure 4), we verified that a strong correlation existed in the case of mix 2 (r = 0.9978, *p*-value < 0.0001) , mix 4 (r = 0.9586, *p*-value = 0.0006) and mix 5 (r = 0.9945, *p*-value < 0.0001). On the contrary, the relationship was not significative for mix 1 (r = 0.6041, *p*-value > 0.05) and mix 3 (r = 0.3631, *p* value > 0.05). This bias could be due to primer competition in a heterogeneous solution, where the final DNA concentration will reflect the initial one when there is no bias in primer affinity for all species; otherwise, the initial and final DNA concentrations would be poorly related. Considering the results reported here, when the analysis rely only on dry weight, the quantitative potential of the technique is partial, but qualitative results (the species list) are reliable. This depends on the fact that same weights do not necessarily carry with them (i) the same number of cells and (ii) the same number of *trn*L copies. On the other hand, when DNA metabarcoding analysis is put in relation to DNA copies, a quantitative correlation can be estimated if primer competition is low or can be measured. In general, a well standardized protocol, from experimental design to bioinformatic analysis, including internal controls and DNA quantification, will maximize the power of DNA metabarcoding.

## 4. Conclusions

In recent years, we face the boosting of new food processes, developed using various tools of science, engineering, and biotechnology. This leads to the placement on the market of food products that are modified, improved, and made to look better, taste different, and be commercially attractive, generating an increase in global trading of such food products and food supplements. Together with the advances in food processing, we are facing also the challenge of new food adulteration and counterfeits strategies and, as a consequence, an accurate authentication is essential (i) for correct labeling, (ii) to ensure food safety, (iii) to protect the consumer, (iv) to sustain the economic value of high quality, controlled, food. At the same time, there is a boosting in molecular techniques development and application, and an exponential increase of molecular data available in GenBank database, paving the way for the overcoming of the caveat and pitfalls demonstrated and discussed in this research.

Molecular authentication has become one of the prospective standards and it is potentially the proper answer to the transformers’ and consumers’ request of quality confidence. Nevertheless any application will require a shared and standardized protocol, which is still lacking despite the fast progresses obtained till now. Today, technical aspects limitations are challenging our effort to obtain an optimal tool for both qualitative and quantitative analyses. Nevertheless, considering the exponential advancement in scientific research and molecular approaches, we are confident that in the next few years we will be able to apply in an extensive, rapid and cost-effective way these molecular tools starting from complex and degraded matrices.

## Figures and Tables

**Figure 1 genes-10-00248-f001:**
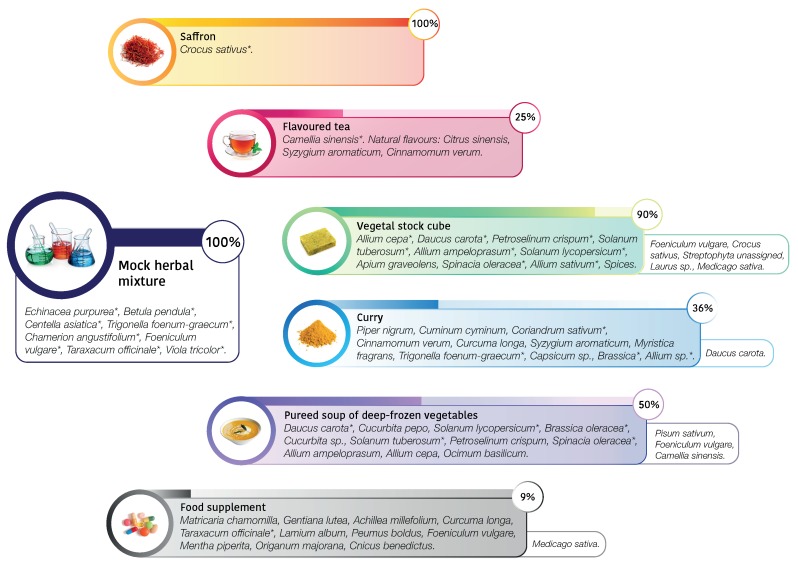
Schematic overview of commercial processed foods and mock herbal mixture tested with DNA metabarcoding approach. The percentage bars report the “observed/expected” percentage, that is the match in what was declared in the ingredients label and what we detected using DNA metabarcoding; not declared specimen/contaminants are excluded from the percentage calculation and listed in the boxes on the right. See also Table 1.

**Figure 2 genes-10-00248-f002:**
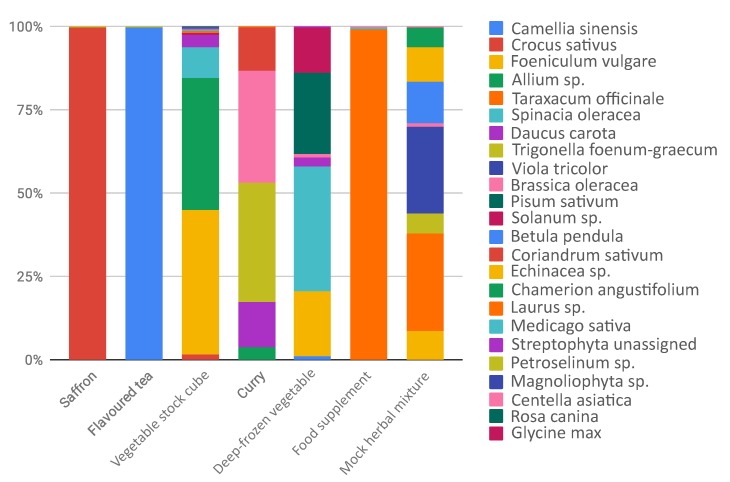
Taxa barplot reporting relative abundances of plants detected in processed food.

**Figure 3 genes-10-00248-f003:**
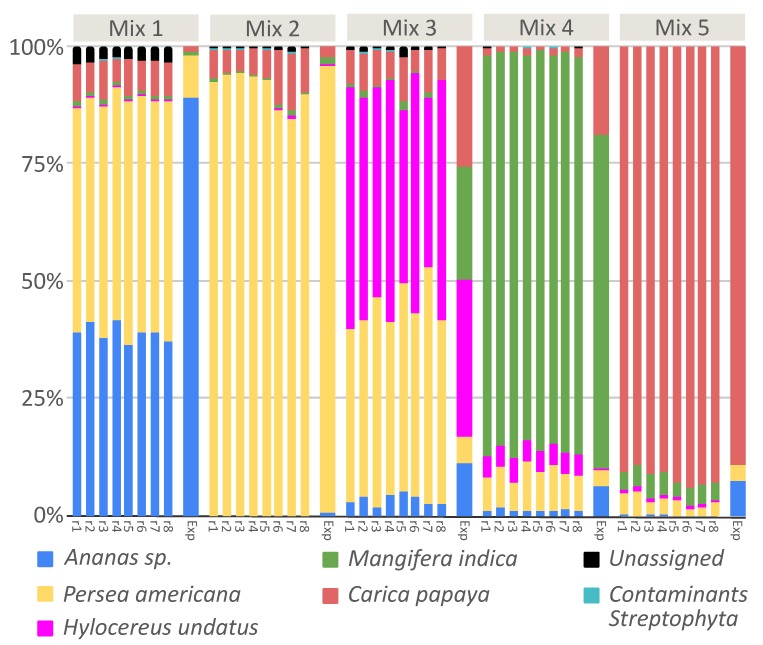
Taxa barplot reporting relative abundances of fruit detected in fruit mixtures, for each replicate. **Exp** represents the expected fruit proportion for each mix.

**Figure 4 genes-10-00248-f004:**
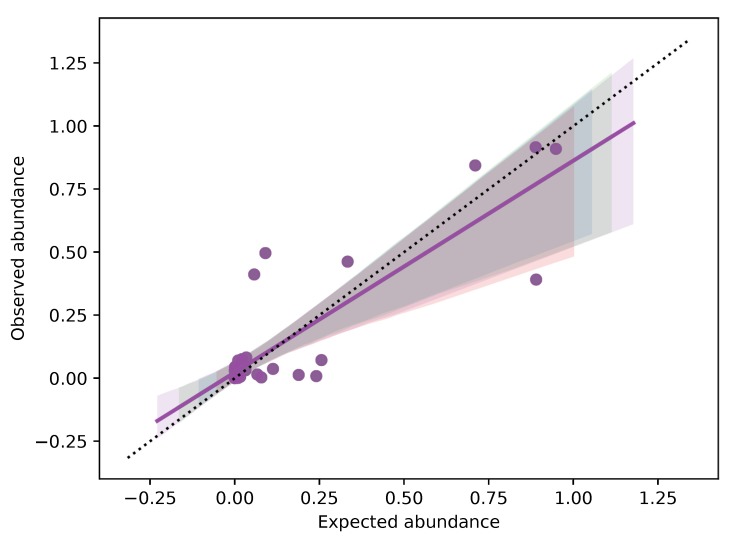
Plots of accuracy and observation correlations for each samples in the five fruit mixtures.

**Table 1 genes-10-00248-t001:** Primer sequences used for Illumina sequencing (*trn*L g-h) and for Sanger sequencing (*trn*L c-h).

*trn*L g	GGGCAATCCTGAGCCAA
*trn*L h	CCATTGAGTCTCTGCACCTATC
*trn*L c	CGAAATCGGTAGACGCTACG

**Table 2 genes-10-00248-t002:** List of processed food and artificial lab mix tested. When percentages were displayed in the ingredients label, the information was reported in the “declared composition” column.

		DNA Metabarcoding	Not Declared Specimen
Sample	Declared Species Composition	Composition (Vsearch)	Contaminants
		(>0.3%)	(False Positive)
saffron	*Crocus sativus*	*Crocus sativus* (99.8%)	
flavoured tea	*Camellia sinensis*	*Camellia sinensis* (99.7%)	
natural flavours:		
*Citrus sinensis*		
*Syzygium aromaticum*		
*Cinnamomum verum*		
vegetable stock cube	*Allium cepa* (1.9%)	*Allium* sp. (39.3%)	*Foeniculum vulgare* (43%)
*Daucus carota* (1%)	*Spinacia oleracea* (9.3%)	*Crocus sativus* (2.1%)
*Petroselinum crispum* (0.5%)	*Daucus carota* (3.9%)	*Streptophyta unassigned* (0.5%)
*Solanum tuberosum* (0.3%)	*Petroselinum* sp. (0.3%)	*Laurus* sp. (0.5%)
*Allium ampeloprasum* (0.2%)	*Solanum* sp. (0.3%)	*Medicago sativa* (0.3%)
*Solanum lycopersicum* (0.1%)		
*Apium graveolens* (0.1%)		
*Spinacia oleracea* (0.1%)		
*Allium sativum* (0.1%)		
spices		
curry	*Piper nigrum*	*Trigonella foenum-graecum* (36%)	*Daucus carota* (13.4%)
*Cuminum cyminum*	*Brassica* sp. (33.3%)	
*Coriandrum sativum*	*Coriandrum sativum* (13.2%)	
*Cinnamomum verum*	*Allium* sp. (4%)	
*Curcuma longa*		
*Syzygium aromaticum*		
*Myristica fragrans*		
*Trigonella foenum-graecum*		
*Capsicum* sp.		
*Brassica* sp.		
*Allium* sp.		
deep-frozen vegetables	*Daucus carota*	*Spinacia oleracea*(37.2%)	*Pisum sativum* (24.8%)
*Cucurbita pepo*	*Solanum* sp. (13.3%)	*Foeniculum vulgare* (19.8%)
*Solanum lycopersicum*	*Daucus carota* (2.6%)	*Camellia sinensis* (1.1%)
*Brassica oleracea*	*Brassica oleracea* (0.9%)	
*Cucurbita* sp.		
*Solanum tuberosum*		
*Petroselinum crispum*		
*Spinacia oleracea*		
*Allium ampeloprasum*		
*Allium cepa*		
*Ocimum basilicum*		
food supplement	*Matricaria chamomilla*	*Taraxacum officinale* (99.2%)	*Medicago sativa* (0.4%)
*Gentiana lutea*		
*Achillea millefolium*		
*Curcuma longa*		
*Taraxacum officinale*		
*Lamium album*		
*Peumus boldus*		
*Foeniculum vulgare*		
*Mentha piperita*		
*Origanum majorana*		
*Cnicus benedictus*		
mock herbal mixture	*Echinacea purpurea*	*Taraxacum officinale* (29.3%)	
*Betula pendula*	*Viola tricolor* (26.3%)	
*Centella asiatica*	*Betula pendula* (12.8%)	
*Trigonella foenum-graecum*	*Echinacea* sp. (9.9%)	
*Chamerion angustifolium*	*Foeniculum vulgare* (8.6%)	
*Foeniculum vulgare*	*Chamerion angustifolium* (6%)	
*Taraxacum officinale*	*Trigonella foenum-graecum*(5.8%)	
*Viola tricolor*	*Centella asiatica* (0.4%)

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
