# Peer review of "Food Tracking Perspective: DNA Metabarcoding to Identify Plant Composition in Complex and Processed Food Products"

_genes, 2019, doi:10.3390/genes10030248_

Round 1
Reviewer 1 Report
Dear authors,
The paper was well written, and very clear especially for readers not very familiar with metabarcoding. I especially like the figures.
Please consider my suggestions below, looking forward to seeing this study published.
Minor comments
Change title to better reflect that the manuscript focuses on plants.
Line 43: proteins not relevant here
Line 46: use hts here and say that it’s sometimes called ngs.
Line 51: maybe write food products, instead of stuff
Line 51-55: please add references for these points
Line 56: incomplete reference databases would lead to false negatives, misidentified reference sequences to false positives.
Line 59: maybe elaborate a bit more on trnL here. Taxonomic resolution, previous studies, especially related to food products.
Line 81: maybe consider adding common names as well.
Line 91: correct me if i misunderstood something here, but i find the adjustments of dna concentration based on trnl qpcr worrying, since u quantify using the same primer that is used for hts, so you don’t really do primer testing. Maybe pooling juice in different proportions and then bulk extracting would have been better.
Line 110: please make sure these reference sequences are available on ncbi.
Line 138: please give the exact size range of sequences retained.
Line 138: ee filtering is an excellent choice, maybe include the citation here since not every one is familiar with this new approach. Edgar & Flyvbjerg 2014
Line 140: this is strange, since usually the most abundant sequence is chosen to represent a cluster. Please explain why a random sequence was used here.
Line 164: quite a short amplicon! Maybe discuss issues of taxonomic resolution, not a problem here since taxa are quite distant, but with other real world samples might contain substitutions with closely related species.
Figure 2: nice figures, all are ready polished and clear actually. Good job
Table 2/line 194: i really like the table.
Line 216: maybe write "reliability of the metabardocing protocol". Would have been good here to have the same taxa in the mock sample that would be expected in the tested sample, to fully confirm primer bias.
Line 238: Feel free to add Elbrecht & Lesse 2015 here https://journals.plos.org/plosone/article?id=10.1371/journal.pone.0130324
Figure 3 Is really cool and clear, good job! Maybe have unassigned sorted to the top of each bar and colour it gray, or back to make it distinct from the other colours.
Figure 4: Consider transforming both axis into Log10, to better see the distribution of low abundant data points.
Please make sure to add a otu table for all samples, including read counts, otu sequences and taxonomy as supporting information
Please provide an accession ID where the raw sequence data can be obtained.
Author Response
The response to the Reviewer 1 is upload as a pdf file :

Reviewer 2 Report
The manuscript written by Bruno et al. describes a DNA metabarcoding approach based on the trnL gh marker to identify the plant composition of several kinds of processed food qualitatively and quantitatively. The authors provide new reference sequences for exotic fruits and simulated herbal and fruit mixtures to test replicability and to assess the quantitative representation of species in mixtures of known DNA content. In my opinion, the manuscript should be considered for publication, because of the combined approach of qualitative and quantitative analysis, whis is rare for this marker. The qualitative results about food composition are within expectations, but I find it very interesting to see the high reproducibility of different PCR reactions.
However, there are some issues that I think need to be addressed in the manuscript.
1. How was cross-contamination monitored and how was the extraction and PCR design? Usually, extraction and PCR negative controls have to be included, especially when using degraded DNA and I think this is especially important in the case of analysing food plants. But I did not see any report of negative controls in this study. Please, mention at which points during the lab works you used negative controls and how they were treated and what you did to prevent contamination in general. Were they sequenced, too? Maybe Crocus in your false positives is a cross-contamination from the lab work? Were samples extracted together in one workflow or on different days? Were PCRs performed together or with dedicated set-ups for each sample?
2. In the text I did not really understand how the mock samples were prepared. It took me a while and the supplement. Maybe you can explain it a little bit clearer in the methods section, that your DNA was combined in certain quantities.
3. How much of the different samples were used for DNA extraction? And moreover the study is lacking a detailed description of the samples. Why did you analyse saffron? What did you expect to find else?
4. The schematic overview of samples is really nice, but to me it is not clear how the food supplement was processed, for example. It would be nice to get more detailed information about what makes the samples worthwhile analysing.
5. If Crocus sativus or Camellia sinensis are the only species detected (Table 1), why are they reported to with 99.8 or 99.7%? What is the rest?
6. I would be cautious with reporting that this marker can be used quantitatively for food composition quality control.
7. I understand why you chose fruits, that have not been processed before in the lab. But overall, I think it would have been better if you chose something more similar to your samples. Then you could maybe find out if the species that you did not detect in the commercial samples are maybe difficult to amplify.
Also, just a small remark from my side: it would have been really
interesting, if your mock mixtures were compared to a DNA extraction
from a juice prepared of equal vs. different proportions of your different fruits. This might give you an indication of what to expect from commercial juice, for example.
Please, find here some more specific critics:
1+18: one of the main goals
52: consistency with lists – here the list uses i) ii), in line 74 it changes to a), b), c) and later back again to i), ii)...
53: change to: ... for which we commissioned a herbalist’s shop and simulated herbal products...
54: grammar: change the sentence for example to: ...the accidental laboratory contamination when the target DNA is fragmented and of low concentration.
57: you can remove the word step after PCR amplification and afterwards start a new paragraph with your objectives. It makes it easier to find them for the reader.
59: include an article before trnL and change to: ...by using the trnL marker. Also include the number of different sample types tested.
89: qPCR used here abbreviated but explanation comes in line 97.
90: I would not write trnL gene, as you amplified and sequenced only the P6-loop of the intron. Maybe you could write: ...and DNA copy numbers of amplified trnL fragments were compared.
98/99: What is the difference between artificial fruit mixtures and fruit mixtures?
109 c) is not part of a list, remove the bracket )
133: For how many reads were you aiming?
138: outside of bounds... doe it mean longer or shorter or both? The marker has a strong length polymorphism.
143: Just a comment: we use the OBITools (Boyer et al. 2016) to create our own trnL reference database always based on the newest embl releases. Maybe this could be convenient for you some day.
174: Figure 1: The text is too small.
184: T. officinal e remove the space before the e
184: Here, my first thought was: have you checked for mismatches in the primer binding sites? Maybe you could put your explanations from 222 here?
186: prossessing: how were the foods processed? The samples are not described detailed enough. Especially the food supplement. Are these pills like in the figure? And what kind of supplement is that? Can you really expect plant DNA from that or are they just taking specific compounds?
200: You were wondering, why some spices are not represented in your sequences. Maybe they are present at low copy number in your DNA and the cycle number was too low? Maybe it is also because seeds might not contain that much chloroplast DNA at all? Maybe seeds contain substances that inhibit PCR or degrade DNA? Have you checked these things?
214: ...as an experimental control...
237: markers
253:HiSeq
259: quantitative -> quantities
268: necessarily
274-277: The sentence is hard to follow and should be re-structured.
293: Table S3: The 3 primer sequences should go into the Methods section.
Author Response
Response to Reviewer 2 are in the upload pdf file.

Round 2
Reviewer 2 Report
Dear authors,
I feel that it really has improved. Please, find here a few last comments:
1 major goals (add the s)
Figure 2 Frozen vegetables and Food supplement should be written without abbreviation and if possible also replace the underscore by a space like this: Streptophyta unassigned
239 +240 sp. not in italics
254 consistency: change T. officinale to Taraxacum officinale
272: maybe change “Assuming the inevitably existence of primer bias” to “Assuming the probability of primer bias”
Best wishes
Author Response
Many thanks to the reviewer for all the useful advices that have allowed us to improve our work.
Responses to comments follow:
1 major goals (add the s)
>Correction done
Figure 2 Frozen vegetables and Food supplement should be written without abbreviation and if possible also replace the underscore by a space like this: Streptophyta unassigned
>Correction done
239 +240 sp. not in italics
>Correction done
254 consistency: change T. officinale to Taraxacum officinale
>Correction done
272: maybe change “Assuming the inevitably existence of primer bias” to “Assuming the probability of primer bias”
>We agree with the reviewer. We changed the sentence in the main text